# Time-Sequential Monitoring of the Early Mesothelial Reaction in the Pleura after Cryoinjury

**DOI:** 10.3390/diagnostics14030292

**Published:** 2024-01-29

**Authors:** Taeyun Kim, Yu-Kyung Chae, Sung-Jin Nam, Haeyoung Lee, Sang-Suk Hwang, Eun-Kee Park, Yeh-Chan Ahn, Chulho Oak

**Affiliations:** 1Department of Internal Medicine, Samsung Medical Center, Seoul 06351, Republic of Korea; jimsb89@naver.com; 2Department of Biomedical Engineering, Pukyong National University, Busan 48513, Republic of Korea; 3Department of Internal Medicine, Kosin University College of Medicine, Busan 49267, Republic of Korea; server28@naver.com (S.-J.N.); hss7056@naver.com (S.-S.H.); 4Department of Thoracic and Cardiovascular Surgery, Kosin University College of Medicine, Busan 46241, Republic of Korea; ansovinus0703@gmail.com; 5Department of Medical Humanities and Social Medicine, Kosin University College of Medicine, Busan 46241, Republic of Korea; ekpark@kosin.ac.kr

**Keywords:** cryoinjury, OCT, mesothelial reaction, pleura

## Abstract

(1) Background: An early mesothelial reaction of the pleura, leading to fibrosis, has been reported in animals after chemical or heavy metal exposure. However, the visual monitoring of early time-sequential mesothelial reaction-associated cryoinjury has not been fully investigated. Therefore, this study aimed to evaluate and visualize the early mesothelial reactions seen following cryoinjury using rabbit pleura. (2) Methods: We monitored the early mesothelial reaction in rabbit pleurae after cryoinjury using optical coherence tomography (OCT), in real-time, which was then compared with pathological images. Due to the penetration limit of OCT, we made a thoracic window to image the parietal and visceral pleurae in vivo. We also used an innovative technique for capturing the microstructure in vivo, employing a computer-controlled intermittent iso-pressure breath hold to reduce respiratory motion, increasing the resolution of OCT. We organized three sample groups: the normal group, the sham group with just a thoracic window, and the experimental group with a thoracic window and cryotherapy. In the experimental group, localized cryoinjury was performed. The mesothelial cells at the level of pleura of the cryotherapy-injured site were visualized by OCT within the first 30 min and then again after 2 days at the same site. (3) Results: In the experimental group, focal thickening of the parietal pleura was observed at the site of cryoinjury using OCT after the first injury, and it was then confirmed pathologically as focal mesothelial cell proliferation. Two days after cryoinjury, diffuse mesothelial cell proliferation in the parietal pleura was noted on the reverse side around the cryoinjured site in the same rabbit. In the sham group, no pleural reaction was found. The OCT and pathological examinations revealed different patterns of mesothelial cell reactions between the parietal and visceral pleurae: the focal proliferation of mesothelial cells was found in the parietal pleura, while only a morphological change from flat cells to cuboidal cells and a thickened monolayer without proliferation of mesothelial cells were found in the visceral pleural. (4) Conclusions: An early mesothelial reaction occurs following cryoinjury to the parietal and visceral pleurae.

## 1. Introduction

Mesothelial cells play a distinctive role in the regeneration of tissues at the injured site. Mesothelial cells share similar characteristics with stem cells that enable proliferation and phenotypic changes when triggered by inflammation or injury, which is called a “mesothelial cell reaction” [1,2,3]. Early mesothelial exposure to foreign substances results in acute mesothelial inflammation. In animal models, this inflammatory process has been induced by heavy metals, talc, and chemicals such as bleomycin [4,5]. Chronic inflammation of the mesothelium in response to asbestos leads to lung diseases such as lung cancer and mesothelioma [6,7].

However, the mechanism of early-stage carcinogenesis in the mesothelium is unclear, and image-based reports have not been published on this topic. In the past, due to anatomical and technological difficulties, only a few attempts have been made to trace the early pleural reaction. Moreover, the area of pleural inflammation could not be specified, nor could it be localized because of poor identification of the injury site. Given that improved imaging techniques to record the effects of asbestos exposure on the pleura would further expand our understanding of the pathologic pathways of pleural inflammation and subsequent cancer development, reproducible and localized modeling of the pleural reaction could facilitate mesothelial imaging.

Cryoinjury has been employed in animal models to address the biological variability associated with the size and location of the injured area, aiming to establish a reproducible animal model [8,9]. The technique can induce localized inflammation of up to 3 or 4 mm in depth, affecting both the parietal and visceral pleurae, as well as the lung parenchyma simultaneously [10]. Previous animal studies have demonstrated that acute lung injury occurs following hepatic cryoablation, leading to a mesothelial cell reaction [11,12]. Notably, cryoinjury offers an advantage over exposing animals to foreign substances for inducing inflammation in the pleural space. While foreign substances can be inserted into the potential pleural space to induce inflammation, they often result in diffuse pleural injury, making the precise determination of the site of the mesothelial reaction challenging. Up to this point, there have not been any studies on mesothelial cell reactions after direct pleural inflammation has been induced by cryoinjury.

Recently, optical coherence tomography (OCT) has been introduced as a non-invasive, high-speed, and high-resolution imaging modality that can provide real-time, cross-sectional images on a micrometer scale. OCT has been widely utilized to visualize various airway injuries. Brenner et al. showed airway swelling due to smoke inhalation [13] and quantified the changes in airway thickness [14]. The same group also detected and monitored airway injury after half mustard gas exposure [15]. These studies were feasible because OCT could visualize the epithelium, mucosa, and cartilage at a high resolution and because the upper airway is easily accessible via endoscopic OCT. In addition, three-dimensional imaging of the chest walls and lungs of rabbits enabled differentiation between normal tissue and chest tumors [16]. In the study, thoracoscopic surgery was performed on rabbits, and a rod OCT probe was inserted into the pleural cavity.

In this context, the present study aims to evaluate the early mesothelial reaction by accessing, visualizing, and histologically investigating the pleura by producing a thoracic window to image the intact pleura, using a benchtop OCT scanner that presents better image quality than any other probe-based scanner. We produced a cryoinjury to the site to induce a mesothelial cell reaction, monitoring the localized reaction and the sequential changes of the mesothelium through a real-time imaging technique, and confirming our findings through pathological correlation.

## 2. Materials and Methods

### 2.1. Animal Preparation

All experiments were performed in accordance with the Guide for the Care and Use of Laboratory Animals. The study protocol was approved by the Committee on Animal Research of the College of Medicine at Kosin University (KMAP-16-11). Twelve male New Zealand white rabbits (Taesung Laboratory Animal Science, Busan, Korea) weighing 3.0 to 3.7 kg, were used for the experiments. The twelve rabbits were divided into the normal control group (*n* = 4), sham-treated group (thoracic window only, *n* = 4), and experimental group (thoracic window + cryoinjury, *n* = 4). The rabbit was euthanized using CO_2_ gas. Initial intramuscular anesthesia was performed with Ketamine 5 mg/kg and Xylazine 0.8 mg/kg. After intubation of the rabbits with a non-cuffed endotracheal tube (3 mm inner diameter, 4.3 mm outer diameter), we maintained anesthetic depth by the injection of 10 mg/h ketamine and 3 mg/h xylazine. The initial ventilator settings were as follows: tidal volume (Vt) 6 mL/kg, frequency 30 breaths/min, positive end-expiratory pressure (PEEP) 0 cmH_2_O. Oxygen saturation was monitored by a pulse oximeter on the ear. The rabbit was fixed to the operation table in a lateral position for preparation and image acquisition. The degrees of mesothelial change were defined as normal, suspicious (hyperemic mucosa without nodular granulation), and abnormal (hyperemic mucosa with nodular granulation), based on the gross findings on day 2 and day 14 after the procedure.

### 2.2. OCT System

Two lab-made spectral-domain OCT systems were used to measure the thickness of the pleura, from the basement membrane to the top of granulation, in real time. We used an 850 nm OCT system with better resolution for imaging the early pleural reaction in the first week, while we used a 1310 nm OCT system with better penetration for imaging the advanced pleural reaction in the second and fourth weeks.

A spectral-domain OCT system, based on the Michelson interferometer, was developed as shown in Figure 1. A broadband light source sent a beam to a fiber-based 2-by-2 beam splitter through an optical fiber and the split beams were then directed to the reference arm and sample arm. A low-coherence light source (Broadlighter D855, Superlum, Cork, Ireland) with a center wavelength of 850 nm or 1310 nm (SLED Butterfly, EXALOS, Schlieren, Switzerland) and a full width at half-maximum of 100 nm was connected to the interferometer. Afterward, the beams were reflected toward the reference mirror and samples, respectively. The reflected beams were coupled into each optical fiber and delivered to the 2-by-2 beam splitter. The merged beam was diffracted by a grating (1800 lines/mm for the 850 nm OCT system or 1145 lines/mm for the 1310 nm OCT system, Wasatch Photonics, Morrisville, NC, USA), depending on wavelength. The interference pattern was measured by a 1 × 4096-line scan camera (Sprint spL4096-140km, Basler, Exton, PA, USA) with a line rate of 140 kHz. A two-axis scanner with a 5 by 5 mm^2^ scanning area was designed and manufactured using two galvanometers (6220H, Cambridge Technology, Bedford, MA, USA). To characterize the OCT system, we measured the point spread function, and it showed a spatial depth resolution of 4 μm in air, a roll-off of 12 dB/mm, and a signal-to-noise ratio of 103 dB.

### 2.3. Thoracic Window

A thoracic window (Figure 2a) was made as described previously [16]. The skin was surgically opened with a 5-centimeter-long transection, and the upper intercostal muscle layers were resected between the third and fourth ribs. The lower layers of muscle tissue were abraded using small forceps, while all muscle fibers overlying the parietal pleura were abscised in a region of approximately 10 × 5 mm^2^. Resection of the lower muscle fibers was carried out very carefully to avoid injury to the pleurae and chest wall and prevent the occurrence of pneumothorax and mechanical stress during preparation.

### 2.4. Cryoinjury to the Pleura

Cryoinjury through a thoracic window was performed under general anesthesia using the iso-pressure breath-hold technique. This technique utilizes a novel breathing pattern of several phases: (1) tidal breathing of 3 mL/kg for 4–5 s, (2) two or three deep breaths, (3) apnea for a period of 10 s with a programmed airway pressure of 20 cmH_2_O, at which time the OCT scanner is allowed to monitor the thoracic window. Figure 3 shows the breathing cycle using the iso-pressure breath-hold technique. The deep breaths prior to the breath-hold recruited the volumes inside the lungs.

We then performed three cycles of −80 °C cryoablation using an Erbokryo cryosurgery unit (ERBE Medizin-Technik GmbH, Tubingen, Germany). A flexible cryoprobe was applied to the parietal pleura, as shown in Figure 2b. The pleura was frozen at −80 °C for 30 s and was then thawed for 30 s. We then repeated cryoablation for two more cycles.

We then performed three cycles of −80 °C cryoablation using an Erbokryo cryosurgery unit (ERBE Medizin-Technik GmbH, Tubingen, Germany). A flexible cryoprobe was applied to the parietal pleura, as shown in Figure 2b. The pleura was frozen at −80 °C for 30 s and then thawed for 30 s. We then repeated cryoablation for two more cycles.

### 2.5. Hematoxylin and Eosin Staining

The chest wall tissue, including the intercostal muscle and ribs, was cut into 6 × 6 cm^2^ pieces, fixed with 10% neutral buffered formalin, and embedded in paraffin. Four-micrometer-thick serial sections were stained with Hematoxylin and Eosin and examined via microscopy.

## 3. Results

### 3.1. Gross Findings of Visceral and Parietal Pleura

Figure 4 shows the gross findings for the pleura and chest wall after 14 days of cryoinjury. The 12 rabbits were sacrificed on day 1 (normal control 4) without cryoinjury, on day 2 (Sham 2, experiment 2), and on day 14 (Sham 2, experiment 2) after cryoinjury. The chest walls of the 12 rabbits were removed on the day of death.

### 3.2. Real-Time In Vivo OCT Image

In vivo OCT imaging was performed to monitor the changes in the mesothelium (Figure 5). The thicknesses were measured from the basement membrane to the top of the mesothelial thickness using OCT. Figure 5 shows one example of real-time sequence images in the experimental group, obtained by OCT through the thoracic window of both the visceral and parietal pleurae after cryoinjury. Figure 5a shows the intact parietal and visceral pleurae. Alveolar space just beneath the visceral pleura was observed. The OCT shows a real-time image of the localized site, where the exudative material could be seen at the site of cryoinjury. Figure 5b–f are OCT images taken 5, 20, 30, and 60 min and 48 h after cryoinjury, showing a progressive focal thickening of the parietal pleura. The extent of focal thickening had spread further along the pleural cavity at every time interval.

### 3.3. Ex Vivo Gross, Histological, and OCT Examination

On day 2 after the procedure, ex vivo samples were harvested, and the mesothelial cell reaction was examined via OCT and histologic study. Upon histological examination, 6 normal lesions (4 in the normal group and 2 in the sham group), 4 suspicious lesions (2 in the sham group and 2 in the experimental group), and 2 abnormal lesions (2 in the experimental group) in the parietal pleura at the site of cryoinjury around the thoracic window were confirmed by a pathologist. Ex vivo images of the parietal pleura in the cryoinjury group on day 2 after cryoinjury are shown in Figure 6.

The OCT image and the corresponding histological images of parietal pleura in the cryoinjury group on the baseline, day 2, and day 14 after cryoinjury are shown in Figure 7. The thickness of the parietal pleura increased as the days passed.

A comparison of the pathological images between the normal pleura and those of the sham group and the cryoinjury group is shown in Figure 8. The images were obtained at baseline and on day 2 and day 14 after cryoinjury. Pathologists confirmed that there was only a mild mesothelial reaction on day 2 in the sham group, while they recorded a mild-to-moderate reaction on day 2 in the cryoinjury group and a moderate-to-severe mesothelial reaction on day 14 in the cryoinjury group.

The mean thicknesses (μm) of the mesothelial reaction of the three groups in the parietal and visceral pleurae were compared in Table 1. The thicknesses of both the visceral and parietal pleurae were highest in the experimental cryoinjury group, followed by the sham group and normal group.

## 4. Discussion

The present study explored the intricate processes of early mesothelial cell reactions in response to cryoinjury, which has not previously been fully investigated. For this study, we accurately approached the pleura via thoracoscopy, by which the accurate placement of OCT was visually guided. With the use of a real-time OCT scanner, we not only observed but also confirmed early mesothelial reactions of the pleurae and correlated these reactions pathologically. Notably, such reactions manifested exclusively in those groups subjected to cryoinjury: the mean thickness of the pleura was significantly greatest at day 14 after cryoinjury. These findings were first visualized by a real-time OCT scanner and then confirmed histologically. One notable aspect of this study is its innovativeness in inducing localized and reproducible microdamage to the pleural mesothelium, tracking early mesothelial reactions for two days through fine OCT imaging, without the need for tissue biopsy. In addition, our findings also suggest the potential role of high-resolution imaging guidance with an OCT scanner in an optical biopsy. Our study might provide further insights in this research field, which, consequently, could lead to elucidating the potential mechanism of carcinogenesis in the pleura. The development of our model in examining early mesothelial reactions in the pleura with the use of an OCT scanner could serve as an opportunity to examine the associations between the markers of early mesothelial reactions, namely, immunohistochemical and molecular biological factors, although further studies are necessary.

A localized mesothelial cell reaction emerged remarkably swiftly, becoming evident as early as 5 min into the OCT observation. In line with a prior investigation in 2007 on mesothelial injury induced by talc, where the levels of interleukin-8, vascular endothelial growth factor, and transforming growth factor-β1 began to rise after 5 to 10 min and the values continuously increased for 6, 24, and 24 h after exposure [17], our findings provide another valuable insight into the temporal dynamics of mesothelial responses using an OCT scanner. Another experimental finding also corroborates our results. Adamson et al. reported a mice study on early responses to acute injury wherein the peak level of mesothelial proliferation occurred within the first 2–4 weeks after asbestos exposure [4]. They concluded that this early mesothelial injury to asbestos could result in the development of the malignant disease, mesothelioma. A loss of control of cell proliferation leads to the formation of neoplastic mesothelial cells [18]. In alignment with these timelines, our study expanded upon early mesothelial cell reactions within the initial 2 days following cryoinjury, validating the findings with the use of a high-resolution OCT scanner and with histological examination.

One of the noteworthy observations of this study involved the focal thickening of the parietal pleura within the first 30 min after cryoinjury. This focal exudative reaction subsequently triggered diffuse thickening around the site of injury within the subsequent 2 days. Consequently, a two-stage model for early mesothelial cell injury might be proposed: an initial focal exudative reaction in the parietal pleura within 30 min post-injury, followed by a subacute mesothelial cell reaction characterized by diffuse mesothelial cell proliferation within 2 days. It is well documented that an exudative inflammatory response occurs in the pleura in response to injuries from infection, inflammation, or neoplastic diseases [19]. In addition, in a model with Crohn’s disease, it was also suggested that early inflammatory exudates might lead to the organization of layers of reactive mesothelial cells. [20]. This early reaction would be followed by subacute mesothelial cell proliferation [7]. A murine model showed that exudate macrophages stimulate the proliferation of mesothelial cells within 2 days [21]. Our results also suggest that the earliest mesothelial reactions might be effectively monitored through an integrated biomarker approach, encompassing both biological and optical biomarkers. The localized points of observation and early in vivo follow-ups provide a nuanced understanding of the temporal aspects of mesothelial reactions, opening avenues for targeted interventions.

Due to the limited resolution available for the examination of pleural structure, elucidating early changes in the mesothelial cells in the pleura has been restricted to the overall acute reaction of the pleural mesothelium after exposure to substances such as talc [22]. In our previous report, we suggested the feasibility of OCT scanner-based measurement in visualizing pleural reactions in response to talc pleurodesis, implicating the potential mechanism in the development of pleural effusion [22]. In that study, a pleural reaction against talc pleurodesis over 4 weeks was assessed by measuring the thickness of pleural reactions using a biophotonic modality spectral domain OCT, which was also used in the current study [22]. We also have shown that an OCT scanner could be useful for evaluating acute and sub-acute pleural reactions. Changes in the pleurae were evaluated using the thickness of the pleura, and the talc pleurodesis group in the 1st and 2nd weeks showed significant thickening of the pleura, while the pleural reaction in the sham and normal groups revealed only minimal changes from their baseline thickness [22]. Another study in 2011 by Muta et al. corroborates those findings [23]: they divided a total of 30 rats into 3 groups (a normal group that received saline, a group that received OK-432, and a group that received talc) and examined the thickness of pleural reaction on day 30 after the exposure in rats. Both the OK-432- and talc-administered groups revealed a significantly higher thickness of the pleura in both microscopic and macroscopic examinations [23]. Thus, both studies have consistently shown that OCT could have a potential role in providing in vivo imaging of the mesothelium, based on thoracoscopic examination and the detection of pleural lesions at an early stage.

Meanwhile, the mesothelial reaction in terms of pleural thickness was more significant in the parietal pleura than in the visceral pleura. This finding was confirmed by both pathological examination and ex vivo OCT measurement: in the pathological examination, the thickness of the parietal pleura of the cryoinjured group was 150 μm and the thickness of the visceral pleura was 30 μm; in an ex vivo OCT image, the thickness of the parietal pleura of the cryoinjured group was 155 μm and the thickness of the visceral pleura was 45 μm. It has also been shown that in the clinical results, mesothelioma and pleural plaques occur exclusively in the parietal mesothelium, and not in the visceral mesothelium [24]. The parietal pleura has traditionally been regarded as the main reference because mesothelioma initiates primarily from the parietal layer. Using principal component analysis and bridge-partial least squares regression models, Røe et al. found in 2010 that parietal and visceral pleural tissues had distinct differential gene expression profiles in patients with mesothelioma [25]. Significant differences in gene expression between the parietal and visceral pleurae were found, and the down-regulation of overexpressed genes in mesothelioma versus normal tissue was found. In another study, no gene ontology entities were overexpressed in the parietal pleura, but several entities were downregulated compared to the visceral pleura, all of which might suggest a propensity for developing mesothelioma, especially in the parietal pleura [26].

The real-time and high-resolution capabilities of OCT are helpful in showing time-sequence events in a delicately changing pleural space in response to cryoinjury. In 2009, Xie et al. first demonstrated the in vivo thoracoscopic imaging capabilities of three-dimensional OCT systems with improved forward-scanning rigid graded-refractive-index lens rod probes in detecting pleural cancer [27]. It was shown that the imaging probe of the system was easily adjustable to suit different locations within the thoracic cavity and that it could be easily modified to other locations, such as for rigid airway endoscopic examinations. In the present study, OCT images showed mesothelial thickening in 6 specimens (2 in the sham group and 4 in the experimental group). Notably, 66% of these instances exhibited suspicious mesothelial cell reactive lesions upon gross examination, underscoring the potential advantages of OCT in imaging the mesothelium for the early detection of pleural lesions during thoracoscopic examination. An OCT system could provide high-resolution images of 1–15 μm with an imaging depth of 2–4 μm. With advances in OCT techniques, the diagnostic applications of OCT could be extended to include obstructive lung disease, lung cancer, airway wall injury due to inhalation, pulmonary hypertension, and pleural diseases [28]. When locating a thoracoscopic biopsy, OCT might be helpful in identifying tumorous regions from non-malignant irregularities, but additional human research is necessary to validate the results obtained from animal models. In addition, as OCT techniques continuously develop, detailed imaging with high resolution could be feasible, and an OCT scanner could guide experimenters to the appropriate site for tissue acquisition. For example, an OCT scanner with precise resolution, enabling the examination of structures within 1 mm, has been increasingly used in the clinical diagnosis of ocular and retinal diseases [29,30].

Increasing levels of air pollution pose a significant threat to public health. The results of our study could further provide the possibility of utilizing OCT-based examination for pleural reactions in response to air pollution. Elevated concentrations of pollutants such as particulate matter and harmful gases could lead to respiratory problems and other adverse health effects. Indeed, there have been growing bodies of interest in and research into the relationship between air pollution and the mesothelium or mesothelial reactions. In a retrospective study in patients with pneumonia with pleural effusion, an increase in nitrogen dioxide was related to a decrease in the differentiation of CD62 and the degrees of pleural molybdenum and zinc, suggesting an association between air pollution exposure and an altered immune response in pleural effusion [31]. Another interesting autopsy-based study in Sao Paulo highlighted lifetime exposure to air pollution as an indicator of pleural anthracosis [32]. Experimental studies support these findings.

## 5. Conclusions

In conclusion, early mesothelial cell reactions in both the parietal and visceral pleurae, which were induced by cryoinjury, were assessed and observed by real-time OCT imaging, and the reactions were histologically confirmed. The findings of our study may contribute to further studies on the biological behavior of the mesothelium in carcinogenesis in the pleura. Future in vivo studies are necessary to confirm our findings in carcinogenesis in the pleura. In addition, further technological developments in the field of imaging processes, such as ultra-high resolution, would improve diagnostic accuracy and the ability to characterize the pathological events that occur following injury to the pleura.

## Figures and Tables

**Figure 1 diagnostics-14-00292-f001:**
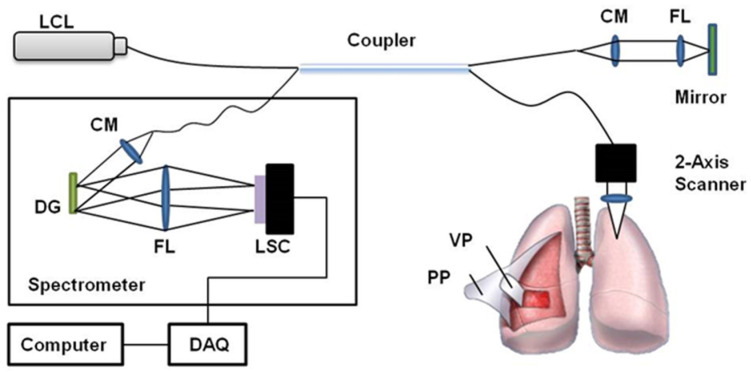
Schematic diagram of the developed spectral-domain OCT. CM: collimator, DG: diffraction grating, FL: focusing lens, LCL: low-coherence light source, LSC: line scan camera, PP: parietal pleura, VP: visceral pleura, DAQ; data acquisition.

**Figure 2 diagnostics-14-00292-f002:**
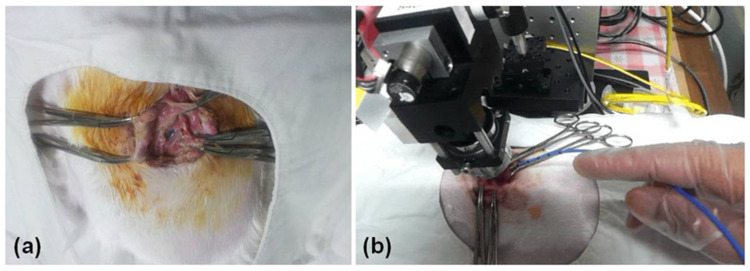
(**a**) Thoracic window, (**b**) cryo-injuring using a flexible cryoprobe under an OCT scanner.

**Figure 3 diagnostics-14-00292-f003:**
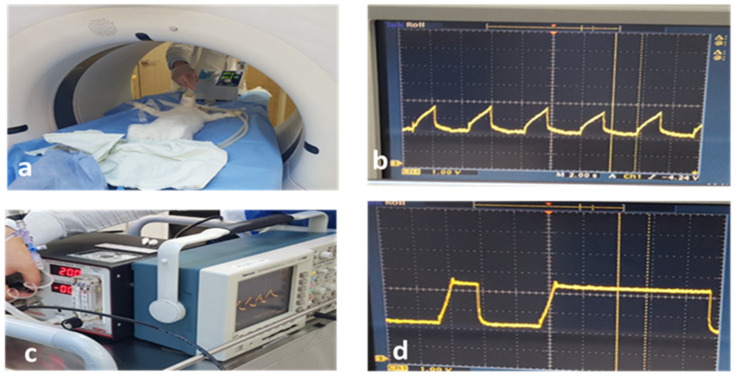
Iso-pressure breath technique. (**a**) Mechanical ventilation, (**b**) tidal breaths, (**c**) controlled breath with fixed airway pressure, and (**d**) breath hold followed by a deep breath.

**Figure 4 diagnostics-14-00292-f004:**
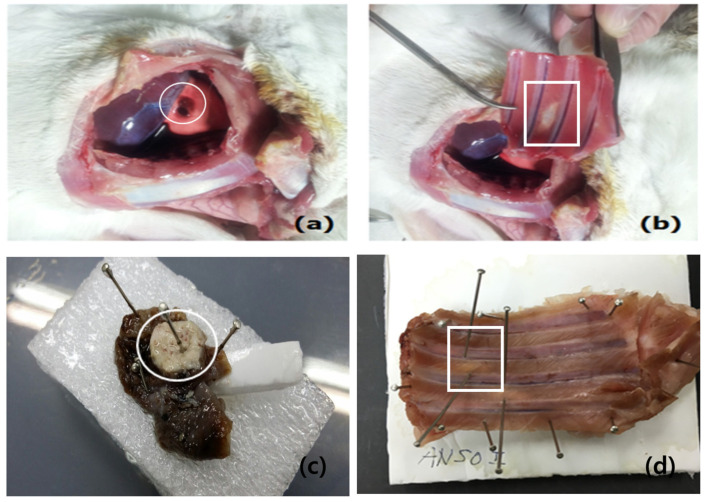
Gross findings of the visceral (**a**,**c**) and parietal (**b**,**d**) pleura on day 14 after cryoinjury. (**a**,**c**) A hyperemic hemorrhagic spot (white circle) on the visceral surface. (**b**,**d**) The thoracic window shows a translucent parietal surface (white square).

**Figure 5 diagnostics-14-00292-f005:**
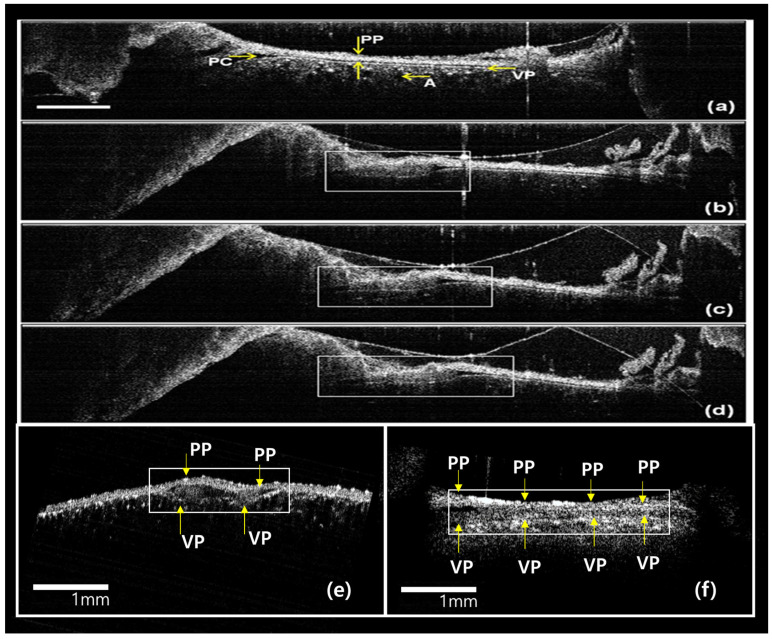
Real-time in vivo OCT images through the thoracic window: (**a**) before cryoinjury, (**b**) 5 min, (**c**) 20 min, (**d**) 30 min, (**e**) 60 min, and (**f**) 48 h after cryoinjury. White boxes show mesothelial cell proliferation in the pleura. A: alveolus, PP: parietal pleura, PC: pleural cavity, VP: visceral pleura. Scale bar: 1 mm.

**Figure 6 diagnostics-14-00292-f006:**
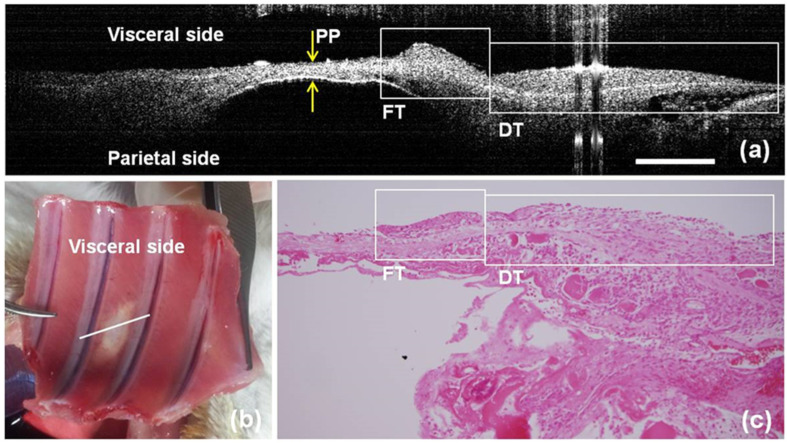
Ex vivo images of the parietal pleura in the cryoinjury group (2 days after cryoinjury). (**a**) OCT image showing focal and diffuse thickening. (**b**) Excised tissue with the thoracic window (the OCT scan area is indicated by a solid line). (**c**) Matched histology showing focal and diffuse proliferations of the mesothelial cells. DT: diffuse thickening, FT: focal thickening, PP: parietal pleura. Scale bar: 500 μm.

**Figure 7 diagnostics-14-00292-f007:**
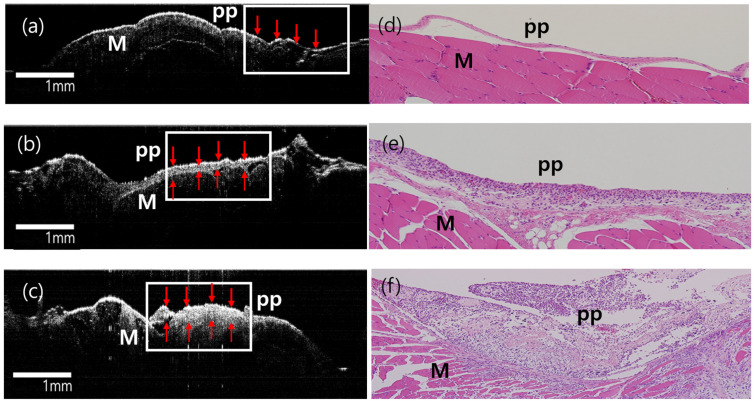
OCT findings according to the degree of early mesothelial reaction after cryoinjury. (**a**–**c**) OCT findings on the baseline, day 2, and day 14 after cryoinjury. (**d**–**f**) The corresponding pathologic findings. (**a**,**d**) OCT image of normal pleura, and the normal pathology (100×). (**b**,**e**) OCT and pathology findings (100×) of mild mesothelial reaction on day 2 after cryoinjury. (**c**,**f**) OCT and pathology findings (100×) of moderate mesothelial reaction on day 14 after cryoinjury. PP: Parietal pleura, M: muscle. Red arrows and white boxes showsthe thickness of the PP increased as the days passed.

**Figure 8 diagnostics-14-00292-f008:**
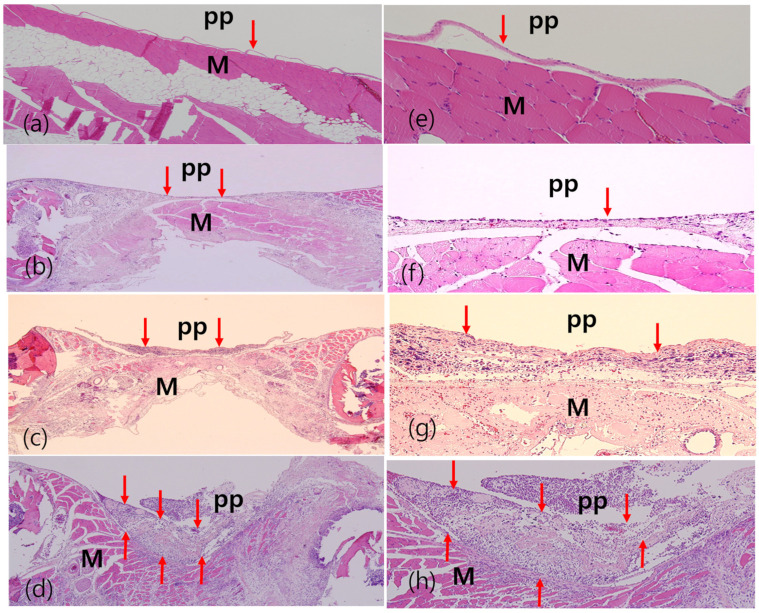
Pathologic findings according to the experimental groups after cryoinjury. (**a**,**e**) Pathologic findings of normal pleura (10×). (**b**,**f**) Pathologic findings (10× and 100×) of the sham group with mild mesothelial reaction on day 2 after cryoinjury. (**c**,**g**) pathological findings (10× and 100×) of cryoinjury group with mild-to-moderate mesothelial reaction on day 2 after cryoinjury. (**d**,**h**) pathological findings (10× and 100×) of moderate-to-severe mesothelial reaction on day 14 after cryoinjury. PP: Parietal pleura, M: muscle. Red arrows indicate the mesothelial reaction.

**Table 1 diagnostics-14-00292-t001:** Pleural thickness, as established by pathology and ex vivo OCT image.

		Parietal Thickening	Visceral Thickening
Pathology	Cryoinjury	150 ± 8.5 μm	30 ± 10.5 μm
	Sham	95 ± 7.6 μm	13 ± 5.5 μm
	Normal	12 ± 3.5 μm	11 ± 2.5 μm
	*p*-Value	0.02	0.04
Ex vivo OCT image	Cryoinjury	155 ± 10.7 μm	45 ± 12.5 μm
	Sham	85 ± 5.5 μm	14 ± 7.5 μm
	Normal	13 ± 2.5 μm	12 ± 3.5 μm
	*p*-Value	0.03	0.04

## Data Availability

Derived data supporting the findings of this study are available from the corresponding author on request.

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
