# Peer review of "Time-Sequential Monitoring of the Early Mesothelial Reaction in the Pleura after Cryoinjury"

_diagnostics, 2024, doi:10.3390/diagnostics14030292_

Round 1
Reviewer 1 Report
Comments and Suggestions for Authors
This is a well designed and well written manuscript. The background summarized very well. The Methods are clearly described and the results briefly summarized. They have discussed their results in a good way and have a good conclusion.
Author Response
We thank you and the reviewer for your interest in our paper and the comments. We would like to thank the reviewer for his or her recommendation.
Reviewer 2 Report
Comments and Suggestions for Authors
Author Response

(The authors gave the same response as above.)

Reviewer 3 Report
Comments and Suggestions for Authors
This is a well written manuscript and an interesting study performed.
I have a few minor comments :
The authors will need to give some background of cryoinjury to the common reader. How is this laboratory work significant and applicable to real life? When does cryo injury occur? Iatrogenic? As a result of real life accident? The common reader would not be able to appreciate the significance of this study. And so some background information is required in the introduction.
Likewise, the discussion section whilst discussing the findings of this study needs to be contrasted to work already performed / literature in this area. This is lacking. How is this work novel and applicable to real life situations? If this is an iatrogenic cause, do physicians or those performing pleuroscopy need to be careful when performing these procedures? Can cryobiopsy be employed in the retrievement of pleura biopsy? How is this finding relevant to the medical community at large? These issues will need to be adequately addressed.
Author Response
We thank you and the reviewers for your interest in our paper and the comments. We have revised the paper in response to these comments and believe it has improved our manuscript substantially. All changes have been marked in red in the revised manuscript. We also provide a point-by-point response to the comments raised by the reviewer.
Comments 1) The authors will need to give some background of cryoinjury to the common reader. How is this laboratory work significant and applicable to real life? When does cryo injury occur? Iatrogenic? As a result of real life accident? The common reader would not be able to appreciate the significance of this study. And so some background information is required in the introduction.
Response 1) Thank you for your valuable comment. We have added and modified sentences in the 3rd paragraph of the Introduction section to enhance readers' understanding of how cryoinjury has been utilized in the research area and why we chose to use cryoinjury for this work. Please refer to the modified 3rd paragraph of the Introduction section.
In addition, we thoroughly reviewed the entire original manuscript and revised to enhance clarity and maintain consistency.
Comment 2) Likewise, the discussion section whilst discussing the findings of this study needs to be contrasted to work already performed / literature in this area. This is lacking. How is this work novel and applicable to real life situations? If this is an iatrogenic cause, do physicians or those performing pleuroscopy need to be careful when performing these procedures? Can cryobiopsy be employed in the retrievement of pleura biopsy? How is this finding relevant to the medical community at large? These issues will need to be adequately addressed.
Response 2) Thank you for comment. First of all, the present study was targeted to construct the early-stage mesothelial reaction model by accessing, visualizing, and histologically investigating the pleura with the use of a benchtop OCT scanner which provides better image quality than any other probe-based scanner. For this, we produced an experimental cryoinjury to the site to induce a mesothelial cell reaction, monitoring the localized reaction and the sequential changes of the mesothelium through real-time OCT scanner image and confirming our findings through pathological correlation. We added and modified sentences discussing the previous literatures dealing with the cryoinjury and addressing the feasibility of this model in a real-world clinical practice. Please refer to the highlighted section in the 3rd, 6th and 7th paragraph of Discussion.